# Towards a Feminist Global Health Policy: Power, intersectionality, and transformation

Hannah Eger [1,2]*, Shubha Chacko [3], Salma El-Gamal [4], Thomas Gerlinger [1], Alexandra Kaasch[2], Marie Meudec[5], Shehnaz Munshi[6,7], Awa Naghipour [8], Emma Rhule [9], Yatirajula Kanaka Sandhya [10], Oriana López Uribe [11,12]

1 School of Public Health, Bielefeld University, Bielefeld, Germany, 2 Faculty of Sociology, Bielefeld University, Bielefeld, Germany, 3 Solidarity Foundation, Bengaluru, Karnataka, India, 4 International Labour Organization, Geneva, Switzerland, 5 Outbreak Research Team, Population Data Hub, Equity & Health Unit, Department of Public Health, Institute of Tropical Medicine, Antwerp, Belgium, 6 School of Public Health, University of the Witwatersrand, Johannesburg, South Africa, 7 University of Limerick, Limerick, Ireland, 8 Department of Sex and Gender Sensitive Medicine, Medical Faculty OWL, Bielefeld University, Bielefeld, Germany, 9 International Institute for Global Health, United Nations University, Kuala Lumpur, Malaysia, 10 The George Institute for Global Health, New Delhi, India, 11 RESURJ, Mexico City, Mexico, 12 Vecinas Feministas, Mexico City, Mexico

☙ These authors contributed equally to this work.
* Hannah.eger@uni-bielefeld.de

**Data Availability Statement:** All data are publicly available and have been submitted as supporting information and/or within the manuscript.

## Abstract

In the realm of global health policy, the intricacies of power dynamics and intersectionality have become increasingly evident. Structurally embedded power hierarchies constitute a significant concern in achieving health for all and demand transformational change. Adopting intersectional feminist approaches potentially mitigates health inequities through more inclusive and responsive health policies. While feminist approaches to foreign and development policies are receiving increasing attention, they are not accorded the importance they deserve in global health policy. This article presents a framework for a Feminist Global Health Policy (FGHP), outlines the objectives and underlying principles and identifies the actors responsible for its meaningful implementation. Recognising that power hierarchies and societal contexts inherently shape research, the proposed framework was developed via a participatory research approach that aligns with feminist principles. Three independent online focus groups were conducted between August and September 2022 with 11 participants affiliated to the global-academic or local-activist level and covering all WHO regions. The qualitative content analysis revealed that a FGHP must be centred on considerations of intersectionality, power and knowledge paradigms to present meaningful alternatives to the current structures. By balancing guiding principles with sensitivity for context-specific adaptations, the framework is designed to be applicable locally and globally, whilst its adoption is intended to advance health equity and reproductive justice, with communities and policymakers identified as the main actors. This study underscores the importance of dismantling power structures by fostering intersectional and participatory approaches for a more equitable global health landscape. The FGHP framework is intended to initiate debate among global health practitioners, policymakers, researchers and communities. Whilst an undeniably intricate and time-consuming process, continuous and collaborative work towards health equity is imperative to translate this vision into practice.

**Funding:** The authors received no specific funding for this work.

**Competing interests:** The authors have declared that no competing interests exist.

## Introduction

In the sphere of global health policy, the imperative to address deeply entrenched discrimination and power imbalances has become increasingly urgent, not least due to their detrimental impact on achieving the Sustainable Development Goals (SDGs) by 2030 [1, 2]. Health inequities, referring to differences in health associated with systemic barriers to good health that are modifiable or avoidable and considered unfair, underscore the need to apply an intersectional feminist lens if transformational change is to happen [3, 4]. Developed through an interdisciplinary and participatory approach, this article presents a transformative Feminist Global Health Policy (FGHP) framework that aligns with intersectional feminist principles, with a particular focus on the interplay between SDGs 3, Health and Wellbeing, 5, Gender Equality, and 10, Reduced Inequalities [1].

An interrogation of power is essential when examining inequitable health outcomes, as its politically, economically and socially induced (mal)distribution in society generates privilege and oppression [5]. Power is most easily recognised as "power-over", which emphasises domination and the existence of hierarchies [6]. Power regimes are the structures through which this dominance is exercised, for example, patriarchy, white supremacy or capitalism. However, different features of power exist, including "power-to" and "power-with" that focus respectively on enabling and collective notions of power [7]. Intersectional feminist approaches advocate for a shift in power towards these more cooperative, less dominant manifestations of power.

Founded on the recognition of the simultaneous existence of multiple interacting factors that determine discrimination and privileges, intersectionality recognises that these factors are not simply additive but rather create complex new structures that in turn require explicit examination to adequately acknowledge the heterogeneity of affected individuals [8–10]. Historically, intersectionality was articulated by Black feminist scholars who recognised the double burden of sexism and racism faced by Black, Indigenous and People of Colour (BIPoC) women in the USA, most prominently by Kimberlé Crenshaw in 1989 [10–14]. By focusing on the most marginalised first, an intersectional approach aims to improve the situation for everyone [11].

### Understanding the intersectional landscape

Intersectionality has proven to be a valuable approach to analyse social structures, shed light on the dominant biomedical frame used in global health that disregards Indigenous and other sources of knowledge, and unpack health inequities [15–20]. To comprehensively address the complexity of health inequities, it is essential to grasp the intricate interplay of power dynamics and interconnected structural determinants such as gender, race and class [12, 13, 21]. These structural determinants, also referred to as social determinants of health (SDOH) inequities by Solar and Irwin [5], influence individuals' and groups' (lack of) opportunities and (in)access to specific resources, leading to structural discrimination of marginalised populations [5, 22]. Structurally embedded power hierarchies that directly impact the global health agenda and individual health outcomes are manifested at the political level, leaving policymakers and political systems with a significant impact on health and wellbeing. For instance, Safaei [23] examined that democracy, grounded in human rights, advances gender equality and improves health outcomes for women.

By considering the multidimensionality of power regimes, including racism, patriarchy and capitalism, an intersectional perspective unveils the deep-seated roots of oppression and the resulting power asymmetries, which in turn lead to the persistence of adverse health outcomes and inequities which are reproduced globally [8, 10, 11, 24, 25].

For instance, the COVID-19 pandemic exposed how intertwined racist and colonial attitudes, merged with capitalist aspirations, resulted in severe vaccine inequities, with the pharmaceutical industry and supporting countries prioritising profits over lives [26–28]. Reproductive justice is a poignant example of the intersectionality inherent in policies that impact sexual and reproductive health and rights (SRHR). Throughout history, interconnected sexist, racist and capitalist aspirations at the political level have impacted the realisation of SRHR [10, 29]. Policies undermining SRHR provide the state with power over an individual's sexuality and bodily autonomy, like in the case of abortion and its widespread criminalisation, the criminalisation of HIV transmission, or several cases of forced sterilisation of Indigenous women [30–32]. Based on an intersectional understanding, reproductive justice considers underlying power regimes and the complexity of SRHR by acknowledging multiple and intertwined layers of discrimination, inequities and lack of access to basic services [29].

## The call for a Feminist Global Health Policy

Despite an increasing body of knowledge on feminist approaches to foreign and development policies [33–35], the global health policy sphere often fails to adequately acknowledge these perspectives. For instance, the enhanced ability of intersectional feminist health research to identify barriers in access to healthcare for women to healthcare that are often overlooked in traditional research and formulate appropriate measures to overcome them has been proven [36]. Highlighting the importance of intersectional approaches to policy, Hankivsky and colleagues [37] developed the Intersectionality-Based Policy Analysis Framework to provide a tool for operationalising intersectionality in health policy analysis for more equitable policies.

This research study aims to explore the contours of a Feminist Global Health Policy (FGHP) and to initiate a debate among global health practitioners, policymakers, communities and researchers. While meaningful information on intersectional feminist approaches is available, the relatively new topic of FGHP currently lacks an action-oriented focus. In order not to remain an abstract construct, concrete and practical instructions are required. Rooted in decoloniality and anti-racism, an intersectional feminist approach to global health policy seeks to dismantle existing hierarchies and challenge entrenched inequities [38–40]. By encompassing holistic, inclusive principles, a FGHP aims to reshape health policies to be responsive to the most marginalised communities and individuals [41] and seeks to advance the transformation of health systems to make them more resilient and better prepared for health risks [42].This research intends to bridge the gap between theoretical understandings and practical implementation of FGHP by examining the research question:

*How can a feminist approach to global health policy be meaningfully implemented to reduce health inequities?*

This study aspires to offer a framework for a FGHP that establishes principles and components of a FGHP, along with guidance for its concrete implementation. The intention is to ensure a link between the strategic, global level and an action-oriented, local level. The unique contribution provided by this research derives from the participatory, intersectional feminist approach. By listening to diverse voices and acknowledging and building on lived experiences, the findings of this study go beyond a literature review and add to the nascent discourse on feminist global health policy. Highlighting the need to challenge existing research and knowledge paradigms, this study attempts to take a first step in this direction, aware of the related challenges and shortcomings.

Spotlighting the need for an intersectional feminist approach, this article underscores the urgency of addressing power imbalances and their impact on health inequities. By examining the complexities of global health policies through the lens of intersectionality, we can pave the

way toward a more equitable and inclusive future. The subsequent sections of this paper will delve into the framework for a FGHP, considering both global principles and sensitivity to local specifications.

## Materials and methods

### Study design

A qualitative study was conducted to answer the research question, provide a political framework for a FGHP, and open the discussion for alternative approaches in global health policy. This research was guided by an interdisciplinary, participatory approach consistent with feminist research principles [43, 44]. Acknowledging that research is not value-free and objective but rather shaped by existing power hierarchies and embedded in a social context [43, 45], feminist research is inherently action-oriented, emphasising participation and empowerment [44, 46, 47]. The choice of research methods is grounded in fundamental principles, i.e. ontology, epistemology and methodology, that reflect on how the world is perceived and knowledge produced [48]. This research was based on a critical, constructionist ontology to underline that existing structures are modifiable [48]. Consequently, this study adopted a postmodern epistemology, rejecting the notion of an objective truth and perceiving reality as socially constructed and subjective [43, 47]. Postmodernism assumes that reality is relative, depending on the social and political context, which further shapes knowledge and experiences [47]. Emphasis is on power relations and their implications [43, 48]. By considering multiple standpoints, a postmodern epistemology is well suited to research intersectionality. In line with postmodernism, this research project followed an interpretivist methodology focusing on *understanding* and considering power and context [48, 49].

Qualitative methods, with their acknowledgement of lived experience and consideration for context and setting, were deemed to be the most aligned with the underlying feminist principles. As such, the qualitative methods applied in this research build on the choice of a constructionist ontology, postmodern epistemology, and interpretivist methodology. To assess the suitability of different methods and value of the research, the work was guided by the eight criteria for qualitative research identified by Tracy [50]: (1) worthy topic, (2) rich rigour, (3) sincerity, (4) credibility, (5) resonance, (6) significant contribution, (7) ethics and (8) meaningful coherence. They were incorporated throughout the research process to allow for this study's maximum possible added value. For instance, to avoid objectifying and exploiting study participants, data for this research project was obtained through focus group discussions (FGDs) in an interactive, collaborative approach, which sought to reduce hierarchies within the research process and centre participants [44, 51, 52].

### Data collection

Three focus groups (FG1, FG2, FG3) with a total of 11 participants were convened between August and September 2022 using the online meetings platform Zoom. Participants were recruited through purposive sampling to obtain the most appropriate composition in relation to the research question, i.e. with regard to geographical diversity and knowledge–through lived experience or work–on relevant topics, including global health, feminism, intersectionality, decoloniality or inequities. Gender was not primarily an inclusion or exclusion factor because a feminist global health policy intends to benefit everyone. However, particular attention was given during the sampling process to ensure that representatives of structurally marginalised groups constituted the majority of respondents. To achieve global representation, participants from all WHO regions attended, with particular emphasis on individuals from across the so-called Global South. Since the FGDs were conducted in English, only English-

speaking participants could be included. Recruitment was undertaken by the principal investigator (HE). The fact that she decided who could participate in this study and the exclusion of people not fluent in English potentially created power hierarchies with implications for the findings, a consideration discussed further in the limitations section.

The recruitment period spanned from 16 June, 2022 to 22 August, 2022. 33 potential participants were contacted via e-mail. Of those, 12 people responded. Due to one no-show, 11 participants (FGD facilitator not included) attended the discussions and meaningfully contributed to the conversations.

The choice of three FGDs was intended to link the global strategic-academic level with the local and action-oriented level. By including different standpoints, the focus was on what is genuinely feasible and required by people on the ground. FG1 comprised six participants affiliated with the strategic and/or academic level related to global topics, such as multilateral organisations, universities and global health institutes. FG2 and FG3 were composed of three and two participants, respectively, who were actively engaged at the local level, including members of civil society organisations, social movements, activists, and marginalised populations. Given the high heterogeneity of the latter, two FGDs were convened with participants active at the local level to enable a richer exchange and a deeper engagement with their respective expertise and experience. Scheduling conflicts due to the large geographical scope also required the convening of three FGDs. Separating the two levels was intentional, as homogenously composed groups are preferred for FGDs [53, 54]. However, this differentiation bears the risk of amplifying socially constructed classifications. The intention was to provide room for both levels, particularly for the local perspective. This approach also allowed for analysing whether the two group types gave different inputs.

The principal investigator facilitated the FGDs, which entailed introducing the research, facilitating participant introductions, and providing discussion prompts. While having conceptualised the FGDs, the principal investigator mainly remained in the background during the discussions, in the sense that she asked the questions and probed to clarify points but not intervene in the conversation. Guiding questions on components, actors and actions regarding a FGHP were developed in advance (see S1 Appendix) to ensure some consistency between FGDs, but with sufficient flexibility to ensure a participant-driven discussion.

## Data analysis

Based on the audio-video recordings, the FGDs were transcribed employing a literal transcription approach, neither phonetic nor summarised (see S2 Appendix). The analysis was conducted according to Kuckartz' [55, 56] qualitative content analysis with the QDA-software MAXQDA. A deductive-inductive code formulation was employed. The data analysis process is visualised in Fig 1. Starting from a deductive approach, emerging codes were added with an inductive approach. As such, the final codes were developed iteratively (see S3 Appendix). When the main and subcodes were considered final, the entire material was recoded with the differentiated code system (Fig 2).

## Reflexivity

To ensure non-exploitative feminist research throughout the process, mechanisms to enhance reflexivity were integrated [43, 57]. A first draft of the results as well as the raw data, including transcripts and video recordings, were sent to the FGD participants, encouraging feedback and requests for changes regarding the analytical process. Member reflection allows for more meaningful and practicable results, which do not solely serve the researcher. Since the results are contextual, this approach ensures that participants agree with the findings and shifts power

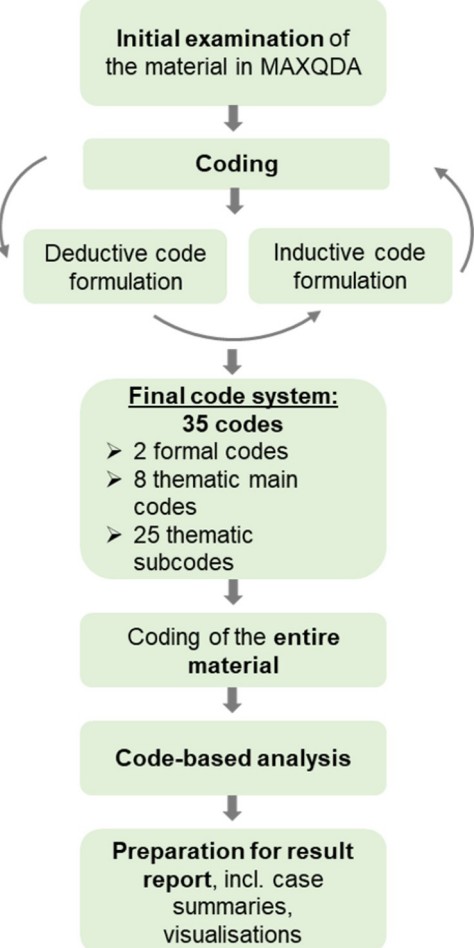

**Fig 1. Data analysis process.**

to them while acknowledging their positionality [50, 58]. The focus group participants were further invited to become co-authors of this article. Therefore, the co-authors of this article are the principal investigator (HE), eight FGD participants (SC, SE, MM, SM, AN, ER, YKS, OLU) and the two study supervisors (AK, TG).

Consistent with postmodernism, the socially constructed reality of the researchers and its influence should be considered, not least to identify persisting power regimes in the research process. The principal researcher and facilitator of the FGDs was a 26-year-old white, middle-class woman from Germany. Her interest in the research topic is strongly motivated by personal and professional experiences and it was part of her master's thesis. She identifies as female, her pronouns are she/her, and society also perceives her as female. She is keenly aware of her highly privileged positions and the consequences of her positionality on this research. Given her prominent role in the research process, power hierarchies persisted despite attempts to reduce and reflect them. In particular, the fact that less participatory approaches were employed in the design of the FGDs and the selection of participants caused an imbalance. Although multiple standpoints were included in this research, most decision-making power concerning implementing the study and the analysis remained with the principal investigator. In accordance with postmodernism, the analysis and findings are subjective and contextualised and should be interpreted accordingly.

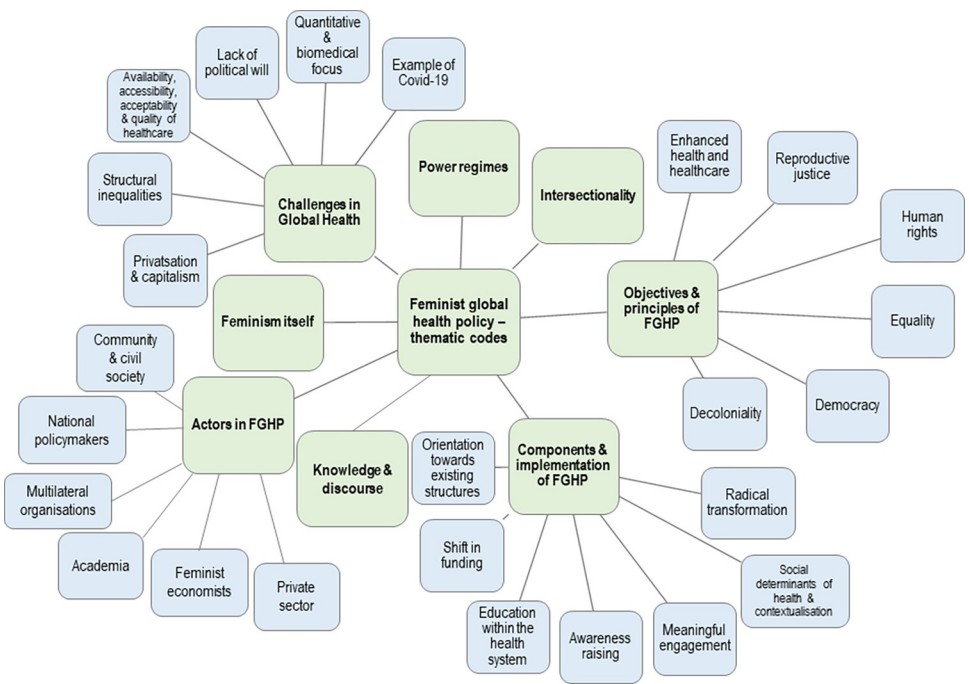

**Fig 2. Visualisation of the code system with thematic main and sub codes.**

### Ethics statement

Ethical approval for this study was obtained from the Ethics Committee of Bielefeld University (Nr. 2022–169). All participants were informed via a Data Protection Information form and a Participant Information form in advance about the procedure, content, purpose, their rights, use of data and data security. Formal consent was obtained in written form, with all informed consent forms signed by the participants before the FGDs.

## Results

Following Kuckartz's [55] method, the qualitative content analysis yielded insights into the development of a feminist global health policy and its implementation. Across the three FGDs, participants mentioned similar aspects of global health and unanimously depicted key challenges and premises for alternative approaches. They emphasised the importance of comprehensively understanding present challenges to develop adequate alternatives. Contemporary structural challenges in global health were discussed, most prominently issues of privatisation and capitalism, structural inequities, problems related to the availability, accessibility, acceptability and quality (AAAQ) of healthcare, lack of political will and a narrow quantitative and biomedical focus. The COVID-19 pandemic was frequently referred to as an example for exhibiting or exacerbating these challenges.

An alternative approach in the form of a FGHP is presented in six thematic blocks that summarise the results of the qualitative analysis and address the challenges identified, including sections on power, intersectionality and knowledge as the underlying considerations that provide the frame for this concept. The identified objectives and guiding principles for an FGHP are then outlined, followed by a depiction of the components that need to be considered in the concrete implementation of this framework. Finally, the relevant stakeholders and their responsibilities in implementing an FGHP are presented.

## Power regimes

All focus groups perceived the role of power to be of utmost importance in analysing existing problems and generating feminist alternatives. Reference to power was made in relation to its negative aspects, i.e., problematic power hierarchies and the unequal distribution of power and its consequences, as well as from a forward-looking perspective, i.e., how existing power regimes can be changed to achieve equity.

Participants criticised the structural power inequities embedded within global health, apparent, for instance, in colonial continuities and global classism. These intersecting inequities arise from power hierarchies that rely on controlling, dehumanising, and feeling superior towards perceived "others", leading to restricted access to healthcare, especially for marginalised groups. The participants further emphasised the gendered dimension of these power imbalances, underscoring that women and non-binary people are particularly affected.

The unequal distribution of power in global health was considered to be closely linked to the privatisation of healthcare and prompted by a capitalist profit orientation, which was referred to as the biggest threat in global health. Concerns were raised about the influence of the private sector and the lack of strong accountability mechanisms:

> *"They have more power than most of our governments."* (FG2)

Unlike governments, the private sector is not accountable to society. Funding decisions do not need to be based on human rights or transparent principles and are, therefore, not necessarily beneficial for patients and communities. The participants depicted how privatisation, resulting from capitalism, intersects with other oppressive power regimes that increase inequities with multiple negative impacts at national and global levels. Capitalistic profit-orientation, in contrast to prioritising health, was portrayed as a threat leading to deleterious health impacts by endangering affordability and equal access to healthcare, whilst neoliberal structures were deemed to exacerbate the exploitation of (female) healthcare workers.

A lack of political will fosters the absence of regulation of the private sector to interfere. This unwillingness to change the system and policymakers' preference for the status quo was a recurring topic in all FGDs, because questioning the current state and intending change would imply a radical transformation. The participants collectively agreed that those in positions of power are generally unwilling to relinquish their power, thus blocking any progress. Privatisation perpetuates this consolidation of power. This seems to be particularly true for authoritarian regimes, which were identified as major threats to health equity:

> *"I think that political leaders who are not held to democratic checks and balances are much more likely to enter into sweetheart deals with corporations [. . .] and so the erosion of democracy and the erosion of political checks and balances [. . .] is so profoundly linked to the kind of corporate capture of healthcare because ultimately it's the decisions of political leaders in countries with weak governance and weak rule of law and rising authoritarianism that permit corporate capture of public goods. So in some ways those two issues of authoritarianism and corporate capture are very linked."* (FG1)

Hence the discussion revealed that the challenges of privatisation and lack of political will are profoundly interrelated with power and uphold structural inequities.

The participants further elaborated that power is related to whose interests are being served, who is included, and who is able to speak. Considering the persistence of power hierarchies, radical change as part of a framework for FGHP was demanded because only by transforming

structures progress can be achieved. A shift in power was regarded as essential in all FGDs. Referring to the remark that *"power inequities are really baked into the global health architecture"* (FG1), it is imperative to first address the global health system. The discussions also included meta-level discourses, questioning whether power is an infinite resource or if shifting power is associated with someone losing power.

## Intersectionality

Intersectionality, as well as power, was an overarching theme that related to both the challenges and the way forward. The participants reflected on multiple intersecting power imbalances that constitute a barrier to healthcare, disproportionately affecting marginalised people. Multiply oppressed individuals are faced with adverse health outcomes, for example due to restricted access to healthcare or a differential treatment based on discrimination. The genuine achievement of universal health coverage, including healthcare that is available, accessible, acceptable and of decent quality, was referred to as an issue requiring an intersectional lens. A holistic approach to address root causes was regarded as essential to eliminate structural inequities and improve health outcomes.

The relevance of intersectionality as a comprehensive frame for a FGHP was frequently emphasised in all discussions and considered pivotal for achieving health equality. To this end, the acknowledgement and inclusion of diverse standpoints at a political level was considered crucial:

*"I do also think that including all genders and all backgrounds in terms of an intersectional approach is absolutely essential. We need to mirror society also in the policy-making processes."* (FG3)

According to the participants, intersectionality enables the recognition of different needs and how to adequately respond to these by developing alternatives. Consistently, a FGHP must be inherently intersectional. The participants highlighted the critical importance of engaging with feminism itself first when dealing with feminist approaches to policy. The colonial and racist history of the feminist movement, which was shaped by predominantly white, Western, middle-class educated women, was recognised as an important concern in the discussions. The impact is still visible today, as the broader movement struggles with power hierarchies and exclusionary visions, practices and mechanisms. The participants suggested internal improvements, including: reflecting on power asymmetries; aligning words with action; improving communication; recognising that plural feminisms exist; and, a genuine embrace of intersectional feminist approaches.

## Knowledge and discourse

Considerations about knowledge and discourse were part of the discussions, including meta-level questions regarding knowledge production as well as more direct approaches to knowledge, discourse and language. It was highlighted that fundamental assumptions must be considered and rethought, since knowledge paradigms are not neutral but shaped by power:

*"Because those who are in power in shaping this discursive power of knowledge, of beliefs, of setting the norms in health are really, you know, unchallenged"* (FG1)

The discussions indicated that this entails thinking outside of common frames in terms of what accounts for knowledge as well as the use of language. Reflections on knowledge and

discourse were perceived as closely associated with colonialism, since existing knowledge paradigms are influenced by a Western perspective, often devaluing other knowledge systems. The participants further outlined that dominant knowledge paradigms favour a narrow quantitative and biomedical focus in global health. The preference for simple, measurable indicators and solutions instead of considering the complexity of SDOH and valuing lived experiences was strongly criticised:

*"Our notions of excellence, and how we recognise excellence are so wedded to one particular way, predominantly kind of an Anglo-European approach or a Western approach of what evidence is, that we discard vast amounts of really important data when we're thinking about effects in health policy"* (FG1)

Furthermore, coloniality determines by whom and how the discourse is shaped. The continuing influence of colonialism was underscored in the discussion, referring to *"the very fact that we're having this conversation in English, yet we are from three different continents"* (FG2). Hence, the analysis revealed that for the conceptualisation and implementation of a FGHP, reflections on who is part of the conversation and listened to are essential.

## Objectives and principles of FGHP

The FGDs addressed overall objectives of FGHP and underlying fundamental principles. A focus on solutions was not deemed appropriate, as this indicates single, easy-to-achieve mechanisms. Instead, the participants preferred the term 'alternatives' to emphasise the complexity and multiplicity of new approaches and challenges.

## Objectives

The two major objectives of FGHP elaborated by the participants were health equity and reproductive justice. Structural change, including power transformation, was perceived as indispensable to achieve these objectives. The participants indicated that health equity implies an improvement of legal and political rights, access to healthcare and enhanced health outcomes. More specifically, universal health coverage is to be achieved, as well as the fulfilment of SRHR. Intersectional feminist approaches were considered valuable and suitable for overcoming current challenges:

*"I could not agree more that feminism takes us from a narrow focus on biomedical solutionism to a broader focus on wellness and health as a complete state of psychological, physical, mental, spiritual wellbeing, and an emphasis on the social determinants of health."* (FG1)

Closely related to health equity, the concept of reproductive justice—encompassing social justice, the importance of the political level, and the relevance of power—was emphasised. Laws and policies tremendously impact SRHR–sometimes with detrimental consequences. To underline this, one participant was referring to the punitive laws in her home country Uganda. At the same time, laws provide an opportunity to accomplish reproductive justice. In this regard, alternative approaches aligning with a FGHP were perceived to advance health equity and reproductive justice.

## Guiding principles

Four fundamental, interconnected principles were considered as essential for FGHP: Human rights, equality, democracy and decoloniality. While contextualisation is of major importance

for FGHP, overarching guiding principles that are globally accepted were deemed necessary by the participants. Such principles represent a common, non-negotiable denominator. However, there may be different approaches on how to achieve these principles, which underlines the pluriversality within global health. It was noted that as with every framework, FGHP has to be flexible and adaptable:

> *"It has to be work in progress [. . .] it's going to be something that allows people to take what they want, add, eject, shape, reshape, you know. [. . .] Like clay where you can keep moulding and re-moulding it."* (FG2)

The principles of human rights, equality, democracy and decoloniality must be acknowledged and accomplished at the political level. This implies reflection and change of current power asymmetries, as well as the active participation of society. Aligning with these principles was regarded as fundamental to address current challenges in global health and advance FGHP. Participants from all FGDs highlighted the importance of human rights, enshrined in the Universal Declaration of Human Rights [59]. Their fulfilment must be at the core of this approach, including the notion of health as a human right. Respecting human rights requires an intersectional lens and is closely related to equality.

The principle of equality is integral when considering health equity. Striving for equality means acknowledging underlying power hierarchies that cause social inequities and acting according to everyone's (heterogenous) needs. The overarching principle of equality is critical for a FGHP to be genuinely holistic and to leave no one behind. Participation and inclusion are integral mechanisms for ensuring equality and also form the basis of democracy.

The significance of democracy for a FGHP was highlighted in the FGDs, as meaningful engagement, the distribution of power and accountability mechanisms are inherently linked to this political system. Since the (national) political context is decisive for adopting a FGHP, democracy is a crucial premise for the substantive implementation of such a policy. It was further noted with concern in the discussions that non-democratic, authoritarian states are more prone to privatisation of healthcare as well as supporting anti-feminist movements. In this respect, the political system of a state impacts healthcare and the health outcomes of its population.

An explicitly intersectional feminist global health policy must feature decoloniality as one of its guiding principles. The discussions underlined the importance of awareness about coloniality, including reflection on intrinsic colonial assumptions. Decoloniality requires tackling persistent power asymmetries and instituting accountability mechanisms to ensure equality, justice and the fulfilment of human rights for everyone. The participants emphasised that advancing decoloniality is a complex task, demanding radical transformation. There is no blueprint for decoloniality; rather, plural approaches were considered necessary.

## Components and implementation of FGHP

Reflecting on concrete implementation, the discussions indicated that emphasis should be on local adaptation, guided by the four guiding principles mentioned above. Based on the recognition that the current system is not functioning and persisting power regimes inherent in societies reproduce inequities, the participants demanded a radical transformation to implement FGHP and to deliver critical improvements in health and equity. The relevance of acknowledging SDOH and local circumstances were strongly emphasised by all participants. Furthermore, all participants agreed that a FGHP can only be successful by involving communities and civil society because they are embedded in and aware of the context:

*"We need to start listening. So we need to stop looking to people in power and who currently hold power and don't have the lived experience of the systems of oppression to provide solutions. Because they actually don't know. [. . .] Yet those who are in those positions of poverty and inequality, they actually know. And so we've got to shift who we listen to."* (FG2)

Meaningful engagement with people on the ground was one of the most discussed topics in the FGDs, with participants emphasising that promoting health equity requires the adoption of participatory approaches.

Contextualisation further entails understanding the political context at every level–locally, nationally and globally. To genuinely shift power, governments must assume the responsibility of ensuring inclusion and participation throughout policy-making processes. Participants emphasised that this is an iterative, continuous process, which is time-consuming, but rewarding and essential. Referring to the 2030 Agenda strapline to leave no one behind [1], the inclusion of those experiencing multiple and intersecting layers of oppression in policy fora was highlighted.

The importance of awareness raising to facilitate social change was a also explored by participants. This includes ensuring communities know their rights to and to promote action to realise those rights. Policymakers likewise need to be aware of good practice methodologies to engage with communities, because this *"would equate to more participation on every level"* (FG3), whilst acknowledging power hierarchies. Participants further discussed the importance of acknowledging reflexivity and positionality, which entails a reflection process about privileges and responsibility. Education and awareness raising within the health system was also discussed as highly relevant, considering harmful power asymmetries that are persistent and continuously reproduced in the health education system and workforce:

*"This sensitisation process in the medical education, where from the beginning, everyone starts to rethink on what are we taught, and who has shaped medicine and medical curricula throughout the ages. Having this critical view upon it, that will also change how we practice medicine in the future ideally [. . .] and in the end, ideally, the patient will profit from it. But I do think that's a very long process."* (FG3)

As highlighted in the quote, a consciousness-expanding process among healthcare personnel was perceived as important to avoid the reproduction of power hierarchies within the health system, as well as sensitisation for the SDOH and contextualised knowledge.

Regarding the challenges related to privatisation and capitalism, a shift in funding was considered essential for a FGHP. Without adequate funding, a framework cannot be meaningfully implemented. The FGD participants demanded that health, as opposed to profits, should be at the centre of funding decisions, which should also align to feminist principles [60–62]. A shift in funding was linked to the transformation of power since money often confers power and incentivises particular behaviours. Again, engagement with community organisations–which are often underfunded–was considered crucial to achieve a meaningful impact. Different mechanisms were proposed in the discussions, including redistributive mechanisms, different taxation systems and more allocation for healthcare and other social sectors. Moreover, multilateral organisations were considered to be important funders if acknowledging the ethical dimension and acting accordingly.

The findings further revealed that present documents and mechanisms could be used as references for FGHP, particularly, the Universal Declaration of Human Rights [59], the Convention on the Elimination of All Forms of Discrimination Against Women [63] and the Beijing

Declaration and Platform for Action [64]. Moreover, national and regional agreements can serve as orientation for FGHP and need to be taken into account for local implementation.

## Actors in FGHP

The focus group participants identified relevant actors for a FGHP and elaborated on their roles and responsibility. Furthermore, questions of accountability were discussed. The participants emphasised that different actors should not be viewed in isolation. Rather, cooperation and participation are crucial for a purposeful FGHP. Community and civil society were regarded as the primary actors for FGHP, the importance of their participation was endorsed in all FGDs, highlighted by this participant:

> *"So for me the number one actor could be community because we are the owners of this framework"* (FG3)

Consistent with an intersectional understanding, community was not perceived as a homogenous group but as a diverse and inclusive set of actors. Community-based and grass-root-level organisations were considered the most essential because of their local presence and context-specific knowledge:

> *"For me it is really community-based organisations, which have or follow feminist ideology and principles. They, I feel, are key. Because they are the ones who are in the community. They have the trust with the community [. . .] So for me, those, they are very key players. Civil society at large yes, but specifically community-based organisations, which have a presence, and subscribe to feminist ideology."* (FG1)

Meaningful engagement of community, and beyond that, the acknowledgement of this sector as a key actor, was associated with improved health care and outcomes by study participants. This also applies to the global context, where inclusion of various communities is important, while acknowledging different backgrounds. In addition to community and civil society, participants emphasised the relevance of national policymakers for FGHP. Encompassing states and governments, as well as individual policymakers, their decision making mandate with respect to policies and laws concerning health, equity, and reproductive justice renders them key actors. Participants of all FGDs identified that accountability for FGHP resides at state level, since *"the states are the duty bearers"* (FG1). Given the many current challenges at the political level, participants demanded systemic change and the transformation of political decision-making.

Besides community and national policymakers, the participants recognised the relevance of multilateral organisations. These organisations are required to develop and monitor universal guidelines and global standards. They are also powerful convenors, with the potential to bring states together and to establish or deepen linkages between the different levels and actors required for a FGHP. Moreover, participants noted that they are influential actors because of their availability of resources, particularly for funding, which again raises important questions about accountability mechanisms.

The discussions identified academia as a secondary actor. As the academic sphere educates the future health workforce, consciousness about its responsibility and mechanisms to restructure the system were considered critical. Academia has the potential to foster change and improve healthcare and health outcomes. The participants implored universities to transform the current neoliberal, quantitative system that neglects meaningful alternatives:

*"Another challenge that I would also like to touch upon is really the entire medical education and the way in which this education is given. So you know the principles of human rights, and all of that are not taught to medical practitioners [. . .] The lack of these skills, which you could call soft skills, are very important, they're critical. But they're not part of the medical education."* (FG1)

Gendered and racist hierarchies within the health workforce are reinforced and upheld by this education system. Acknowledging inherent power asymmetries and acting adequately and in collaboration was considered necessary.

Funding mechanisms must be transformed and adjusted to align with the principles of FGHP. In this respect, feminist economists and funders were introduced as relevant additional actors. Feminist economists advocate for different perspectives on health economies that contrast with the current capitalistic system and its exploitative, oppressing characteristics. Regarding FGHP, the private sector was not considered an integral actor, since it was overwhelmingly associated with challenges in the global health system. However, the participants emphasised that private sector actors must be held accountable and acknowledge their responsibility and–often detrimental–consequences of their actions.

## Discussion

This research represents a major effort to develop a comprehensive framework for a feminist global health policy by drawing insights from diversified lived experiences and expertise and addressing major contemporary challenges in global health. Taking into account diverse standpoints and experiences and recognising their subjectivity, the aggregated results of the FGDs constitute the foundation of this framework. While the framework addresses global considerations, it is not designed to be a definitive, ready-to-use universal product but an initial guide. Moreover, it was repeatedly emphasised in the FGDs that it must be a dynamic and flexible instrument, which provides guidance for further engagement with the topic and can be adapted to specific contexts and conditions. The results in the form of a FGHP framework seek to inform the contours of such a global policy and what needs to be considered in its implementation, rather than providing a straightforward, immediately applicable tool–because the utility of such a universal instrument is questionable and certainly can and should not be developed within one research study. Fig 3 synthesises the proposed FGHP framework that visualises the underlying principles and core components.

The analysis revealed that critical considerations of power regimes, intersectionality and knowledge paradigms must be integrated into the framework (Fig 3). These elements are perceived as pivotal to addressing the complex and interconnected issues of global health and promoting genuine health equity and reproductive justice for all individuals. At the heart of the framework lies a set of fundamental, globally applicable guiding principles, encompassing human rights, equality, democracy and decoloniality (Fig 3). By centring these principles, FGHP seeks to challenge prevailing power structures and historical oppressions that have perpetuated health disparities and inequities worldwide. However, implementing a FGHP must be context-specific and accommodate the respective SDOH that uniquely shape health outcomes in different regions and communities. The findings identified meaningful engagement of community and awareness raising at all levels as pivotal components of a FGHP (Fig 3). Empowering communities and ensuring their involvement in decision-making can lead to more targeted and effective health interventions tailored to their specific needs. At the same time, awareness raising fosters understanding and empathy across diverse populations.

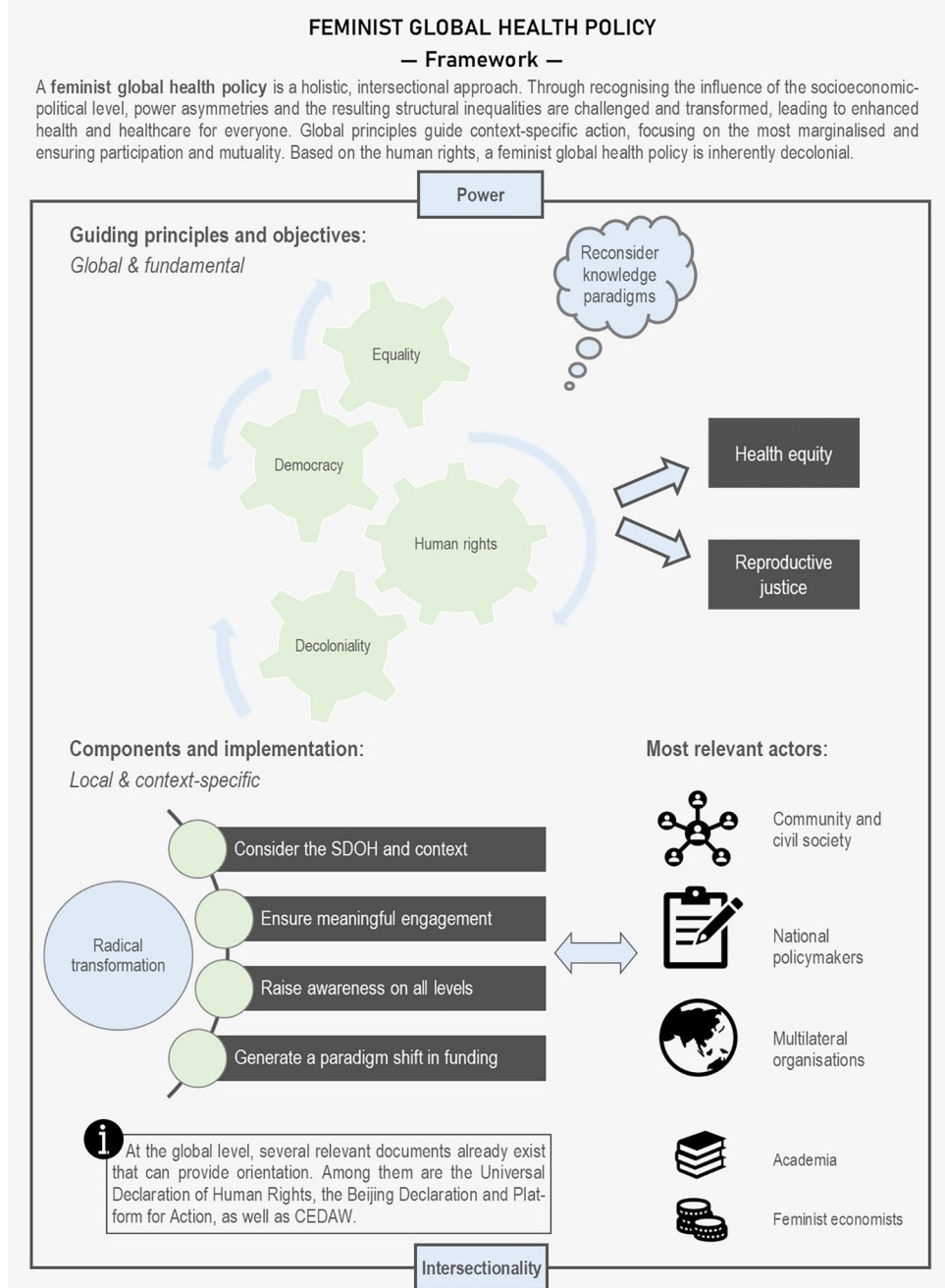

**Fig 3. Synthesis of a feminist global health policy.**

One of the most transformative aspects of FGHP is the call for a profound shift in funding priorities. It advocates a departure from the current profit-driven orientation of the private sector towards feminist approaches. By redirecting financial resources in support of feminist principles and projects, the aim is to ensure the financial feasibility of FGHP and promote equitable access to healthcare services. Adopting and implementing a FGHP denotes a radical transformation of the status quo, challenging deeply entrenched norms and practices that perpetuate health inequities. Key actors in this process are communities and civil society, whose

lived experiences and voices are central to shaping health policies. National policymakers play a critical role in enacting legislative measures that align with the principles of FGHP and in ensuring meaningful engagement. Moreover, multilateral organisations, particularly United Nations institutions, are vital in promoting and funding FGHP initiatives globally. As secondary actors, academia and feminist economists contribute to sustaining a feminist approach to global health policy through research, advocacy and funding (Fig 3).

The alignment of the identified principles and components of a FGHP with existing international agreements, most prominently the Universal Declaration of Human Rights [59] and CEDAW [63], as well as the Beijing Declaration and Platform for Action [64], highlight the framework's compatibility with globally recognised standards for human rights and gender equity. Despite limitations due to the use of exclusively quantitative indicators to measure progress on the SDGs, the objectives of the 2030 Agenda and in particular of SDGs 3, 5, and 10 (on health, gender equality, and reduced inequalities respectively) also conform to the findings and underscore the relevance of FGHP in advancing the broader development agenda [1]. Hence, these globally established agreements provide orientation when adopting a FGHP. The findings align with the existing literature on this topic, as briefly presented below.

Feminist approaches have become increasingly prominent in foreign policy and so-called development policy in recent years. As these policy fields are closely interlinked with global health policy, intersectoral exchange and coherence between these policy domains should be fostered. Feminist global health policy is still in its infancy, but it shares common ground with more established feminist policy fields. For example, in the Centre for feminist foreign policy (CFFP) manifesto for a feminist foreign policy, their approach is outlined as genuinely intersectional, decolonial, transformative, and human-rights centred [65]. Close parallels also exist with publications by German civil society organisations about feminist development policy that put power dynamics and the inclusion of communities at the centre [66, 67]. Accordingly, the FGHP framework presented here closely relates to existing feminist approaches to foreign policy and global cooperation policy [65–67]. This reinforces the relevance of the current findings and the potential to align a FGHP with other feminist policy approaches. It remains to be seen–with hope–whether feminist global health policy will be the next policy field to enter the mainstream debate.

However, it is important to consider and reflect on the criticism of feminist approaches to policy, to avoid these shortcomings afflicting a FGHP from the onset. Several interlinked issues are frequently raised in the literature. The most prominent concern is the risk of feminist approaches being co-opted by mainstream politics and neoliberal structures, leading to a dilution and depoliticisation of feminism. Critics argue that feminism is being used in a tokenistic way by policymakers to benefit their agenda and image, thereby disregarding the original intention of dismantling power structures [34, 35, 68–70]. By integrating feminist approaches into existing structures, power imbalances are perpetuated, not challenged [35, 68–71]. Feminist approaches adopted by governments often fail to adequately acknowledge multiple oppressions and neglect intersectionality as well as the plurality of feminisms [34, 68–70, 72]. In this regard, feminist policies, particularly as designed and implemented by Global North countries, are criticised for not adequately advancing decolonial and postcolonial approaches [69, 70]. Feminist policies should avoid being perceived as a Western concept that is being imposed elsewhere [68, 71]. This includes acknowledging local and contextual specificities, including different forms and shapes of feminisms. For example, regarding feminist foreign policy as a global project threatens silencing small, impactful grassroots movements that may not call themselves feminist but which share the same underlying principles [68, 70, 71].

The understanding of feminist policies as a dynamic concept is crucial. Adaptation to the local context must be at the forefront, while both, the local and global spheres should be

considered. Philipson Garcia and Velasco [33] discuss how these spheres are tightly interlinked and should be perceived as a "local-global continuum" rather than a binary concept, especially with regard to feminist foreign policy. Existing feminist policies are furthermore criticised for not being sufficiently implemented and lacking translation from rhetoric to action [72]. With regard to a FGHP, these risks should be considered and adequately countered.

This study identified foundational principles and components that, if adhered to, can avoid the pitfalls outlined above. Integrating and living up to these principles will ideally avoid the co-optation, dilution and depoliticisation of FGHP. To move beyond political lip service and enable structural change, feminist approaches must be sufficiently funded, owned by the people on the ground and be implemented across all sectors [33, 72, 73]. Indeed, civil society and communities were regarded as pivotal for the implementation and referred to as "owners of this framework" (FG3). This approach further reflects the notion of a global-local continuum. In ensuring a participatory, transformative process reflecting specific needs and circumstances, a FGHP, as a global overarching framework, can be adapted locally to best serve local needs. Furthermore, while it is important to consider contexual adaptations at each level, they should not be viewed in silos since they influence each other. The global health architecture, which is shaped by a colonial and patriarchal mindset, influences people's health as well as their (social, political, economic) environment. The FGHP framework presented intends to demonstrate this interconnectedness.

In principle, this study considers the essential components needed to provide a meaningful contribution. However, the impact of this framework depends on its comprehensive implementation which cannot be assessed in advance. Conceptualising and adapting a FGHP is an ongoing process that will require constant critical reflection. Literature specifically on feminist global health policy shares considerable similarities to the results of the present study. This reinforces the potential impacts of the FGHP framework. For instance, publications by the CFFP [39] and Davies and colleagues [40] closely correspond to the reported findings of this research, due to the adherence to intersectional feminist principles. In addition to explicitly feminist policies, the results identified can be compared to evidence on gender and health equity more broadly, as feminist approaches to global health are not necessarily always labelled as such.

The link between human rights and equity and their significance for global health is well established and was emphasised in the FGDs. Safaei [23] confirms the correlation between human rights, democracy and women's health. Equity is also a prerequisite for the progressive decolonisation of global health. The findings from the FGDs correspond to the positions of Abimbola and Pai [74] and Büyüm and colleagues [75] by indicating that achieving genuine decoloniality requires a systemic transformation.

Meaningfully engaging communities and ensuring accountability at the political level were regarded in the FGDs as pivotal for FGHP. This perspective is reaffirmed by multiple studies [e.g. 76]. Heymann and colleagues [77] examined the importance of social participation and raising awareness to dismantle power relations and restrictive gender norms, as a means to reduce gender inequality and improve health outcomes. Concordantly, Gupta and colleagues [78] identified civil society organisations as crucial actors–when adequately and sustainably funded. The FGD participants strongly emphasised the need for feminist funding structures and sustainable financing if significant impact is to be had.

Gupta and colleagues [78] also advocate for enhanced accountability mechanisms. Transparency and accountability are furthermore regarded as essential to decolonise global health [79]. The critical importance of the political level for health (equity) is frequently noted, due to its role in the development and adoption of laws and policies and the inherent potential to enhance health and gender equity, at national as well as global level [77]. In this regard, multilateral organisations, in particular UN institutions, are considered as central actors and

funders. However, consistent with the presented findings, less dependence of those organisations on external actors and more global authority is required as well as the application of feminist principles to economic decisions [62, 80, 81].

The existing literature often provides more detailed recommendations about specific aspects of the health system. The objective of this research was to develop an overarching global framework that could be applied to multiple areas of concern, i.e. a framework that elaborates a set of cross-cutting principles and components, which when implemented could lead to improvements across multiple sectors and contexts. Due to this global and strategic scope, examples were not provided or discussed in detail in the FGDs. While some publications specifically focus on narrower research topics, they resonate with the broader framework for FGHP. This includes for instance research about intersectionality in healthcare systems [82], reproductive justice [83], gender equity in the health workforce [78, 84] or representation at leadership level [77, 85]. Accordingly, these studies constitute a coherent continuation of the aforementioned findings. This cohesiveness demonstrates the potential positive impacts of FGHP when applied to various areas of health policy. The significance of climate justice and planetary health for a FGHP is one priority area for which further in-depth examination is essential, considering that the climate crisis constitutes one of the major contemporary challenges for global health and human development [86, 87].

## Strengths and limitations

As with any research endeavour, this study has its strengths and limitations. The inclusion of multiple standpoints in the FGDs presents a major strength and reflects a participatory and intersectional approach by acknowledging interlinked determinants, privileges, and oppression [88–90]. Considering coloniality in global health and the urgent need for transformation, participatory research in the form of focus groups is highly relevant [91]. According to the WHO categorisation [92], this research can count as gender-transformative, since power regimes and resulting structural (gendered) inequities were addressed in the FGDs and are challenged in the framework. Moreover, this research is intended to foster transformative change in global health policy. The application of feminist methods using intersectional lenses facilitated a decolonial, participatory approach, providing a valuable space for diverse voices to be heard. The appropriateness of focus groups as a data collection instrument was also highlighted by the participants themselves; the dynamic flexibility of focus groups allowed for nuanced insights and vivid discussions, which would not necessarily have emerged in individual interviews [54, 93]. Online focus groups in particular feature a higher level of accessibility: participants could attend irrespective of their geographical location without the need to travel and almost free of costs. It is a great added value of this study to have involved and connected people and perspectives from different regions of the world to discuss FGHP.

However, some limitations of this study must be acknowledged. The scope of the participatory approach was by necessity limited to the data collection and dissemination of results, rather than involving participants in the entire research process. To fully realise the potential of participation and reduce power hierarchies, future research may explore more comprehensive participatory research approaches, without overstretching the resources of–and thus exploiting–the participants [94]. Finding a balance between genuine participation and not demanding too much energy, time and effort from the participants was a challenge of this research and a compromise to this end was sought. Regarding the principal researcher's positionality and her role as facilitator of the FGDs, not all power hierarchies could be dismantled during the research process, even if the decision for focus groups and a participatory approach reduced power asymmetries. This strongly influences the findings.

Some specific limitations refer to the conduct of focus groups. The non-response rate for focus groups is considerably high and was two-thirds in this study. The inclusion and exclusion criteria also reflect some limitations. The focus groups had to proceed in English (as the contemporary international lingua franca), to ensure communication and mutual understanding between participants. However, this can be regarded as a shortcoming with respect to perpetuating coloniality in global health. Individuals not proficient in English at a high level were excluded. Such an approach risks inadvertently excluding or bypassing meaningful contributions of community members and people with lived experiences. Consequently, the discussions and thus the results were shaped by people adept at talking about the topics examined. While the study participants also spoke on behalf of structurally excluded people and communities and took their perspectives into account by being feminist activists or belonging to community-based NGOs, marginalised people were not directly represented in the majority in this study. By not adequately including these voices, not all power imbalances within the study could be eliminated. In addition, only those with access to a stable internet connection could attend, which further excluded potential participants [93]. Connectivity issues also caused difficulties during the focus groups, including problems related to power outages and non-attendance without notice. The online format of the focus groups, while enabling broader participation, may have affected interaction and the establishment of a familiar atmosphere among participants [93]. Multiple affiliations of participants further complicated the organisation and allocation of the focus groups. This emphasises the complexity of the social world as well as intersecting identities. The context and positionality of the FGD participants could have been highlighted more throughout the research. This implies that a different group composition could have resulted in slightly divergent outcomes [54]. It also raises the questions of how many focus groups and participants are adequate, considering that saturation is not an anticipated criterion in a postmodern perspective. Rather, this research can be understood as providing insights for a continuing process that is not completed with this work.

## Conclusion

In conclusion, this research project constitutes a comprehensive, interdisciplinary effort to elaborate a framework for feminist global health policy. It was guided by the idea–and ideal–of a feminist approach towards global health policy, a somewhat utopian and aspirational vision which provides a space for imagination and reflection. Inspired by bell hooks' words "What we cannot imagine cannot come into being." [95], this study intended to stimulate and provoke reconsiderations of power dynamics and epistemologies. This research offers decisive added value as the foundation of the framework informed by considerations of intersectionality, power relations and knowledge paradigms. By placing power at the centre of the FGHP framework, a comprehensive understanding of intersectionality and the structural roots of inequities is fostered. The embodiment of an intersectional approach highlights both the interconnectedness of inequities and the importance of collaboration between social movements to enable radical transformation.

As with all research projects, several questions remained unanswered in this study. Principles and components for a meaningful implementation of a FGHP were identified, however, power must shift for sustainable change. This aspect requires further considerations. Knowledge and ideas about implementing policies already exist, but genuine action is still lacking. While this study provides important recommendations, further action-based research on contextual implementation is needed to turn the imagination space into action. Shifting power also requires sensitisation of those already in power about the significance of transferring power for the benefit of all people. How can people in power be reached, educated, and convinced? How to proceed in non-democratic contexts? Changing power systems further demands intensive reflections on power itself, as well as on further definitions and concepts,

e.g. global health and feminism. Established definitions often do not reflect the influence of power regimes, mainly coloniality and patriarchy, that is embedded in knowledge paradigms. In light of the emergence of anti-feminist movements around the world, questions arise as to how to strengthen feminist movements? How can a space of collaboration and allyship for the common good be created that entails different intentions and interests?

Policymakers and researchers are encouraged to adopt and test the framework, with future research focusing on its application in diverse settings and exploring its potential for flexibility and adaptation. Practical recommendations deriving from this study are provided in Table 1. By committing to an intersectional, decolonial approach, stakeholders in global health can work together to challenge existing power dynamics and realise the vision of a more equitable and inclusive world. This implies that one universal solution does not exist. Those in power

**Table 1. Recommendations for policymakers and researchers.**

| Recommendations | | |
|---|---|---|
| **Policymaking** | | **Research** |
| **Multilateral** | **National** | |
| ➢ Implement mechanisms to reflect on privileges (at individual & institutional level) | | ➢ Practice reflexivity: Reflect on positionality and the effects on research |
| ➢ Reflect and address power regimes (at institutional & societal level) to enable a shift in power:<br>○ Meaningfully include community & civil society in all processes<br>○ Apply participatory, collaborative approaches | | ➢ Use the presented framework for future research<br>○ Extend or adjust the framework<br>○ Research the feasibility of the framework, e.g. implementation for a specific context or topic |
| ➢ Engage with new approaches that transcend the status quo, including adopting intersectional feminist principles and recognise that feminism is not synonymous with women | | ➢ Apply research tools that can be used to study intersectionality in its complexity, do not rely on an exclusively quantitative approach |
| ➢ Centre on intersectionality:<br>○ Acknowledge multiple power asymmetries and resulting implications for health<br>○ Do not neglect marginalised people, include communities | | ➢ Apply intersectional feminist research methods and reduce power imbalances in the research process through constant reflection and power analysis |
| ➢ Encourage intersectoral work & the Health in all policies approach | | ➢ Appreciate lived experience and other sources of evidence to resist the neoliberal approach often preferred in academic research |
| ➢ Prioritise achieving health equity & reproductive justice by focusing on structural inequalities and the SDOH | | |
| ➢ Restrict the influence of the private sector & profit orientation in health significantly | | ➢ Conduct participatory research: |
| ➢ Reinforce globally established agreements | ➢ Fulfil democratic values and acknowledge accountability to the population | ○ Research for the people affected and not for yourself / your career<br>○ Include the community at all stages, engage with the people meaningfully<br>○ Needs-based, decolonial orientation |
| ➢ Further elaborate globally applicable guidelines for FGHP based on the presented framework | ➢ Adhere to global agreements, including globally agreed guidelines for FGHP | ➢ In the academic context: |
| ➢ Provide space for international exchange | ➢ Transfer agreements into action through meaningful implement-tation | ○ Adjust the curriculum for health workers to overcome the manifestation of harmful power hierarchies in the health workforce<br>○ Teach in accordance with a FGHP |
| ➢ Operate as watchdog, ensure effective monitoring and accountability mechanisms | ➢ Ensure healthcare is affordable, accessible, available & of high quality | |
| ➢ Strengthen the UN and related organisations, particularly regarding more (financial) independence & authority | ➢ Ensure adequate renumeration and decent working conditions for the health workforce and eliminate gender & other inequities therein | |
| ➢ Advance decolonial approaches | ➢ Deal with cases of discrimination in healthcare appropriately and sensitively | |
| ➢ Reinforce the WHO to live up to its mandate and the promise of health for all, including health equity and reproductive justice | | |
| ➢ Reconsider global spending for health<br>○ Advance global redistribution mechanisms<br>○ Reject the colonial "donor-aid" mentality | | |

must reflect on knowledge paradigms and power dynamics. Discouragement among policy-makers can be avoided by embarking on this process together with the community and continuous awareness raising at all levels. Listening to and involving marginalised peoples enables a shift in mindsets and power towards better health for all.

Intersectional feminist global health policy requires a paradigm shift. Albeit being an intricate and time-consuming process, continuous and collaborative work towards the vision of health equity is imperative to translate it into practice.

## Supporting information

**S1 Appendix. Guiding questions for facilitating the FGDs.**
(PDF)

**S2 Appendix. Transcripts of the FGDs.**
(PDF)

**S3 Appendix. Code system.**
(PDF)

## Acknowledgments

We wish to acknowledge and thank Charisse Jordan, Deborah Leticia Akumu, Jonathan Cohen and Victoria Saint for the time and crucial insights that they shared with us during this research.

## Author Contributions

**Conceptualization:** Hannah Eger.

**Data curation:** Hannah Eger.

**Formal analysis:** Hannah Eger.

**Methodology:** Hannah Eger.

**Project administration:** Hannah Eger.

**Resources:** Hannah Eger.

**Supervision:** Thomas Gerlinger, Alexandra Kaasch.

**Visualization:** Hannah Eger.

**Writing – original draft:** Hannah Eger, Shubha Chacko, Salma El-Gamal, Marie Meudec, Shehnaz Munshi, Awa Naghipour, Emma Rhule, Yatirajula Kanaka Sandhya, Oriana López Uribe.

**Writing – review & editing:** Hannah Eger, Shubha Chacko, Salma El-Gamal, Thomas Gerlinger, Alexandra Kaasch, Marie Meudec, Shehnaz Munshi, Awa Naghipour, Emma Rhule, Yatirajula Kanaka Sandhya, Oriana López Uribe.

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
