## [Decision Letter · Decision Letter 0]

21 Nov 2023

PGPH-D-23-01948

Towards a Feminist Global Health Policy: power, intersectionality, and transformation

Dear Dr. Eger,

Thank you for submitting your manuscript to PLOS Global Public Health. After careful consideration, we feel that it has merit but does not fully meet PLOS Global Public Health’s publication criteria as it currently stands. Therefore, we invite you to submit a revised version of the manuscript that addresses the points raised during the review process.

We look forward to receiving your revised manuscript.

Kind regards,

Sreeparna Chattopadhyay, Phd

Academic Editor

Journal Requirements:

1. In the online submission form, you indicated that "All data used in this article can be made available upon request and after consultation with the focus group participants who are not anonymised in the material. A high degree of transparency and traceability is ensured by an extensive methods section.". All PLOS journals now require all data underlying the findings described in their manuscript to be freely available to other researchers, either 1. In a public repository, 2. Within the manuscript itself, or 3. Uploaded as supplementary information.

2. Some material included in your submission may be copyrighted. According to PLOS’s copyright policy, authors who use figures or other material (e.g., graphics, clipart, maps) from another author or copyright holder must demonstrate or obtain permission to publish this material under the Creative Commons Attribution 4.0 International (CC BY 4.0) License used by PLOS journals. Please closely review the details of PLOS’s copyright requirements here: PLOS Licenses and Copyright. If you need to request permissions from a copyright holder, you may use PLOS's Copyright Content Permission form.

Potential Copyright Issues:

Fig 3: Please confirm whether you drew the images / clip-art within the figure panels by hand. If you did not draw the images, please provide (a) a link to the source of the images or icons and their license / terms of use; or (b) written permission from the copyright holder to publish the images or icons under our CC-BY 4.0 license. Alternatively, you may replace the images with open source alternatives. See these open source resources you may use to replace images / clip-art:

- https://openclipart.org/

Additional Editor Comments (if provided):

Dear Authors,

Thank you for making your submission to PLOS Global Public Health. The decision on your manuscript is a major revision. The reviewers have painstakingly reviewed your manuscript and offered constructive feedback for improvement. Please incorporate these before your resubmit it to PLOS Global Public Health.

Regards,

Sreeparna Chattopadhyay, PhD. [Academic Editor, PLOS Global Public Health]

Reviewer # 1

Thanks for the opportunity to review the paper. I appreciate the authors' contribution to the field of global health and their exploration of the intersectionality feminism in the context of global health policies. The topic is both interesting and important, providing a valuable perspective on an area that warrants further exploration. The authors connect their arguments to the global health need, emphasizing the relevance and urgency of a proposed feminist global health policy.

One of the strengths of the manuscript lies in its ability to bring to fore the importance of integrating intersectional feminism into global health discourse. However, I recommend that the authors consider revising the manuscript to provide a more in-depth overview of existing literature and efforts related to intersectional feminism in health. It is essential to delineate how their approach differs from or adds value to the existing body of knowledge. As it stands, it is unclear the added value of the small number of FGDs to the existing literature and frameworks, and what emerged from these FGDs that could have not been captured by a synthesis of the substantive amount of work already done in the field. This could enhance the manuscript's clarity and help readers understand the unique contributions being made.

The proposed GHFP is presented to have been based on data from three FGDs with total of 11 participants, who are recruited through purposive sampling. FGDs are only conducted in English, which indicates that participants were having a rather advanced level of fluency of and comfort with English language, further indicating an advanced level of literacy for non-native English speakers, which already introduces some biases in the sample. The authors indicate that they had one FGD with individuals from academia and international organizations, and another two with people from civils society, grassroot and local levels. Why weren’t participants randomly mixed? This intentional differentiation risks creating hierarchies of knowledge. The authors mention the limitation of the process, but does not reflect more on the influence on the outcome.

There are further no details about the FGDs guides and how the conversations were guided. It would be important to share the guiding questions and provide further information how these were developed. Further, having a facilitator that ‘remained in the background’ seems an oxymoron. It is indicated that the lead researcher, a young white European woman with an advanced university degree was leading the conversation. How was she facilitating the FGDs yet remained in the background? Why weren’t someone external recruited for facilitating the discussion? There is an inherent power dynamic between the ‘researcher’ who is at the same time facilitator, and those who are participating, layered by other intersectional factors playing a role. The reflexivity section lacks any mention about above these aspects. Though these are considered in the limitation section of the paper. Perhaps the sections on reflexivity and limitation should be revisited to consider what should go where, and whether they could be combined.

Furthermore, reflexivity needs to also be applied in the process of developing the guiding questions, the selection of the participants, coding and analytical processes. Transcripts and audio recordings are not the ‘findings’, but the raw data. Content analysis is by nature a subjective process. Were the participants invited to provide input into the analytical process and the ‘findings’ following coding and analysis?

In addition, while the authors declare the ‘positionality’ of the lead researcher, stating that “the analysis and findings are subjective and contextualised and should be interpreted accordingly”, the framework is presented as a ‘global health policy’ framework, something that can be perceived to serve as a universal one. This needs to be further discussed and elaborated.

The findings are summarizing some of the points, yet does not sufficiently contextualize the statements. For example, “The absence of regulation is fostered by a lack of political will to

interfere.” (252). As the participants are based in different countries, it seems that the governments and the political will may be highly different in different countries. This is a bit too general conclusion.

Furthermore, the manuscript would benefit from examination of the critiques against feminist foreign policies and feminist development policies to date. Addressing these critiques is crucial to understanding the potential challenges the proposed feminist global health policy might face and how their new propose framework intends to tackle these concerns. Specifically, the authors should explore whether their approach sufficiently avoids the pitfalls of co-optation, dilution, and depoliticization of feminism. In the current global health landscape, which is often entrenched in colonial and patriarchal structures, it is essential to critically evaluate whether the proposed policy can indeed navigate these challenges effectively in the existing structures and institutions which are often rather hierarchical and patriarchal. Addressing these aspects would strengthen the manuscript and provide valuable insights for policymakers and practitioners looking to implement similar approaches.

In summary, while the manuscript presents an interesting perspective on intersectional feminism in global health, but a more comprehensive description of the methodology, discussion of existing literature, critiques, and potential strategies to avoid co-optation is necessary. Furthermore, I advise that the title is changed to better reflect what the paper presents, I believe that with these revisions, the manuscript could significantly enhance its contribution to the field.

Reviewer #2

Dear authors, thank you for this important and timely piece of work. As it stands, the manuscript shows some merit to be considered for publication but requires heavy editing and major revisions in various aspects to be ready for publication in peer reviewed journals, such as PLoS GPH.

Overarching comments:

Some specific points/concepts/words seem sprinkled in the manuscript without a clear purpose. For instance, feminism, decoloniality, intersectionality, and equity are closely linked. However, for a peer-reviewed scientific paper (and not an editorial/opinion piece), you need to make sure what is the point you want to highlight. If you’re focusing on intersectional feminism, you must be intentional about using points on indigenous knowledge and elaborate on why you brought that point into the paper.

Concepts of health, healthcare, equity, and equality have been conflated throughout this manuscript. They must be clearly defined, and using them interchangeably must be avoided.

While it is clear the authors intend to contribute to the discourse, they should refrain from making bold claims that can’t be directly linked to the focus groups; they should explain what they exactly mean by concepts such as health equality or the fulfillment of bodily autonomy (bodily autonomy is more of a concept to consider and center, not a binary condition easily achieved, and so the wording when using such concepts must be very carefully examined).

Often, it was unclear how the authors moved from point A to point B and drew connections between quotes from the FGs and presented conclusions.

The introduction section is a bit of patchwork, repeating exact keywords without a strong and cohesive flow to the points presented, missing a significant opportunity to present a strong argument for the need for this study.

Overall, the manuscript needs to be edited for language, accuracy, cohesion and flow. I would also encourage the authors to consider what the exact objective of the study was, why were the FGs convened, and continue to present those points. As it stands, this manuscript is elevating certain important points, but does not really introduce a framework, nor tools to utilize/apply that framework. I believe this should be considered throughout the revision.

Line 57: It is important to define intersectionality and, of course, important to note its roots in black feminist legal activism

Lines 66-67: The mention of GBV is abrupt and not followed up in the paragraph.

Lines 76-79: Need references to the claims/points made.

Lines 79-82: It would be more helpful if you elaborate on how these power regimes come into play, how intersectionality plays out, how these intertwined layers operate, and explain your points to set the scene for the reader.

Lines 83-84: This sentence is unnecessarily complex. Please rewrite to make your points clear.

Line 85 (and point repeated in Line 132): You introduce SDoH, but you have used political and structural DoH before this point. It is essential to 1. not conflate these concepts with each other, and 2. clearly define whichever one you want to use.

Lines 86-87: While correct, the point on indigenous knowledge seems to be mentioned suddenly and without proper introduction and follow-up.

Lines 88 & 107: Decolonial might be a better word instead of decolonizing/decolonization

Lines 99-100: Use appropriate references to highlight the increasing calls.

Line 126: Needs a reference.

Lines 132-133: This sentence is unnecessarily complex. Please rewrite to make your points clear.

Lines 140-141: Unclear what this sentence adds.

Lines 143-145: “Drawing on constructionist ontology, postmodern epistemology, and interpretivist methodology, we used qualitative methods in this research.”

I recommend starting the methods section with the most important piece of information: this is a qualitative study. Then bring up the abovementioned frameworks to elaborate on how they are applied to qualitative data interpretation.

Line 155: Unclear what you mean by “specific criteria for qualitative research were perceived as appropriate. “Maybe mention the “the following specific criteria.”

Line 153: It is unclear how this section pertains to quality. And what quality? Quality of the overall project? Data quality? Data collection methods quality? Participant’s quality? Their selection criteria quality?

Line 161: Convened instead of conducted.

Line 162: Online meetings platform, Zoom

Line 163: what is the most appropriate composition? You need to elaborate on this appropriateness and its criteria.

Line 167: What do you mean by Majority World, given it has no reference yet is capitalized similarly to a particular name or concept?

Lines 170-172: How was this done and achieved?

Line 182: Elaborate how feminist principles come into play here.

Line 185: A vibrant discussion… is not a scientific term and does not add to this paragraph either.

Data collection section: It is unclear how FG2 and 3 differed. The composition of FGs is not mentioned. It is not clear how many people were in each FG. How the FGs convened in terms of moderation and/or data collection is unclear. What questions, themes, or discussion topics were introduced? How long did each FG meet for? How many times? You need to explicitly mention the gender, level of seniority, region, nationality, educational background, etc., of participants, in aggregate, to paint a clear picture of whose collective contributions led to the following sections while maintaining anonymity.

Figure 1: An arrow moving from deductive/inductive coding to the final code system is missing.

Figure 2: You shouldn’t use abbreviations in figures without introducing them. AAAQ is an abbreviation.

Lines 204-205: Did they? I think this needs to be elaborated.

Lines 212-213: I’m unsure about including this in the positionality and reflexivity statement.

Reflexivity section: It is important to include other co-authors. If they are all FG members, this needs to be explicitly mentioned. If not, their positionality has also influenced the write-up of this research and is worth noting in the section.

Line 230: Figure 3 should come further in the manuscript, perhaps in the discussion alongside recommendations/Table 1, and should be explained and elaborated.

Line 291: Intersectionality

This section can be expanded on much more, given that it is a central theme in the manuscript title as well, but as it stands, it’s not at all informative, nor does it express how intersectionality is relevant to the solutions put forth by the authors.

Line 322: “Feminism itself”. I believe the use of the word “itself” makes it sound conversational. “Feminism” should suffice to convey the message here. If changed, this should be fixed in Figures as well.

Line 332: Objectives and principles section: see overall comments at the beginning.

This section starts by introducing the four principles but moves on not to dive deep into any of the principles; instead, it continues presenting results. It is a very piecemeal section, very long, and the absence of subheadings makes it very difficult to follow and understand.

Line 333: FGD is being used for the first time here.

Line 355: Democracy has not been discussed before here, except for one quote in line 256. If it is one of the four principles, it must be addressed much earlier in the paper.

Lines 367-380: A lot of repetition and not very cohesive. Language should be heavily edited to highlight the most important points and remove repetitions.

Line 387 & 526: UN Sustainable Development Goals 2030 (add reference)

Line 389-390: What do you mean by multiple oppressed individuals? As it stands now, it reads as multiple individuals, i.e., the plurality and number are important. I believe you intend to emphasize individuals experiencing multiple and intersecting layers of oppression.

Line 391: Beginning of what? The policy-making process?

Lines 415-417: Another example of repetitive points that are not really coherent or placed well in the section.

Line 475-476: This paragraph is about academia, yet in the middle of the paragraph you add a very strong point about the health workforce without further elaboration.

Line 482: This claim needs references to some of these feminist economic theories or practices.

Line 548-549: The authors must be very cautious when drawing a conclusion about the robustness and validity of a concept. As this manuscript is submitted for peer review and conducted according to a specific qualitative methodology, validity should not be claimed based on similarities with other published work that have not been conducted following similar or other scientific methodology. The CFFP paper is a desk review type of publication, and the Davies et al. paper is a viewpoint. Both are great additions to the discourse but cannot be used in a point about the validity of a framework introduced in this paper.

Line 566: FGD is used again.

Line 583: Respected or implemented?

Line 600: Terminate? Or transform?

Lines 631-632: The non-response must be elaborated. Does this mean the FGs were convened but participants did not attend? Does the number 11 include all who were invited to FGs, or all who showed up but didn’t contribute to the conversation? Or all who participated and contributed to the conversation? This is a significant point you need to explain.

Lines 657-658: Deeply embedded or informed by?

Line 683: It would be great to elaborate on the recommendations and briefly describe where each recommendation is drawn from - this could happen earlier in the discussion section. For example: A, B, and C, raised by the FG and the literature (add references), led us to make this recommendation: X

Table 1: Recommendations are not actionable, e.g., Adopt more holistic approaches within global health research.

Table 1: Research column, second point: extend, not extent.

Reviewers' comments:

Reviewer's Responses to Questions

**Comments to the Author**

1. Does this manuscript meet PLOS Global Public Health’s publication criteria? Is the manuscript technically sound, and do the data support the conclusions? The manuscript must describe methodologically and ethically rigorous research with conclusions that are appropriately drawn based on the data presented.

Reviewer #1: Partly

Reviewer #2: Partly

2. Has the statistical analysis been performed appropriately and rigorously?

Reviewer #1: N/A

Reviewer #2: N/A

3. Have the authors made all data underlying the findings in their manuscript fully available (please refer to the Data Availability Statement at the start of the manuscript PDF file)?

Reviewer #1: Yes

Reviewer #2: Yes

4. Is the manuscript presented in an intelligible fashion and written in standard English?

Reviewer #1: Yes

Reviewer #2: No

5. Review Comments to the Author

Reviewer #1: Thanks for the opportunity to review the paper. I appreciate the authors' contribution to the field of global health and their exploration of the intersectionality feminism in the context of global health policies. The topic is both interesting and important, providing a valuable perspective on an area that warrants further exploration. The authors connect their arguments to the global health need, emphasizing the relevance and urgency of a proposed feminist global health policy.

One of the strengths of the manuscript lies in its ability to bring to fore the importance of integrating intersectional feminism into global health discourse. However, I recommend that the authors consider revising the manuscript to provide a more in-depth overview of existing literature and efforts related to intersectional feminism in health. It is essential to delineate how their approach differs from or adds value to the existing body of knowledge. As it stands, it is unclear the added value of the small number of FGDs to the existing literature and frameworks, and what emerged from these FGDs that could have not been captured by a synthesis of the substantive amount of work already done in the field. This could enhance the manuscript's clarity and help readers understand the unique contributions being made.

The proposed GHFP is presented to have been based on data from three FGDs with total of 11 participants, who are recruited through purposive sampling. FGDs are only conducted in English, which indicates that participants were having a rather advanced level of fluency of and comfort with English language, further indicating an advanced level of literacy for non-native English speakers, which already introduces some biases in the sample. The authors indicate that they had one FGD with individuals from academia and international organizations, and another two with people from civils society, grassroot and local levels. Why weren’t participants randomly mixed? This intentional differentiation risks creating hierarchies of knowledge. The authors mention the limitation of the process, but does not reflect more on the influence on the outcome.

There are further no details about the FGDs guides and how the conversations were guided. It would be important to share the guiding questions and provide further information how these were developed. Further, having a facilitator that ‘remained in the background’ seems an oxymoron. It is indicated that the lead researcher, a young white European woman with an advanced university degree was leading the conversation. How was she facilitating the FGDs yet remained in the background? Why weren’t someone external recruited for facilitating the discussion? There is an inherent power dynamic between the ‘researcher’ who is at the same time facilitator, and those who are participating, layered by other intersectional factors playing a role. The reflexivity section lacks any mention about above these aspects. Though these are considered in the limitation section of the paper. Perhaps the sections on reflexivity and limitation should be revisited to consider what should go where, and whether they could be combined.

Furthermore, reflexivity needs to also be applied in the process of developing the guiding questions, the selection of the participants, coding and analytical processes. Transcripts and audio recordings are not the ‘findings’, but the raw data. Content analysis is by nature a subjective process. Were the participants invited to provide input into the analytical process and the ‘findings’ following coding and analysis?

In addition, while the authors declare the ‘positionality’ of the lead researcher, stating that “the analysis and findings are subjective and contextualised and should be interpreted accordingly”, the framework is presented as a ‘global health policy’ framework, something that can be perceived to serve as a universal one. This needs to be further discussed and elaborated.

The findings are summarizing some of the points, yet does not sufficiently contextualize the statements. For example, “The absence of regulation is fostered by a lack of political will to

interfere.” (252). As the participants are based in different countries, it seems that the governments and the political will may be highly different in different countries. This is a bit too general conclusion.

Furthermore, the manuscript would benefit from examination of the critiques against feminist foreign policies and feminist development policies to date. Addressing these critiques is crucial to understanding the potential challenges the proposed feminist global health policy might face and how their new propose framework intends to tackle these concerns. Specifically, the authors should explore whether their approach sufficiently avoids the pitfalls of co-optation, dilution, and depoliticization of feminism. In the current global health landscape, which is often entrenched in colonial and patriarchal structures, it is essential to critically evaluate whether the proposed policy can indeed navigate these challenges effectively in the existing structures and institutions which are often rather hierarchical and patriarchal. Addressing these aspects would strengthen the manuscript and provide valuable insights for policymakers and practitioners looking to implement similar approaches.

In summary, while the manuscript presents an interesting perspective on intersectional feminism in global health, but a more comprehensive description of the methodology, discussion of existing literature, critiques, and potential strategies to avoid co-optation is necessary. Furthermore, I advise that the title is changed to better reflect what the paper presents, I believe that with these revisions, the manuscript could significantly enhance its contribution to the field.

Reviewer #2: Dear authors, thank you for this important and timely piece of work. As it stands, the manuscript shows some merit to be considered for publication but requires heavy editing and major revisions in various aspects to be ready for publication in peer reviewed journals, such as PLoS GPH.

Overarching comments:

Some specific points/concepts/words seem sprinkled in the manuscript without a clear purpose. For instance, feminism, decoloniality, intersectionality, and equity are closely linked. However, for a peer-reviewed scientific paper (and not an editorial/opinion piece), you need to make sure what is the point you want to highlight. If you’re focusing on intersectional feminism, you must be intentional about using points on indigenous knowledge and elaborate on why you brought that point into the paper.

Concepts of health, healthcare, equity, and equality have been conflated throughout this manuscript. They must be clearly defined, and using them interchangeably must be avoided.

While it is clear the authors intend to contribute to the discourse, they should refrain from making bold claims that can’t be directly linked to the focus groups; they should explain what they exactly mean by concepts such as health equality or the fulfillment of bodily autonomy (bodily autonomy is more of a concept to consider and center, not a binary condition easily achieved, and so the wording when using such concepts must be very carefully examined).

Often, it was unclear how the authors moved from point A to point B and drew connections between quotes from the FGs and presented conclusions.

The introduction section is a bit of patchwork, repeating exact keywords without a strong and cohesive flow to the points presented, missing a significant opportunity to present a strong argument for the need for this study.

Overall, the manuscript needs to be edited for language, accuracy, cohesion and flow. I would also encourage the authors to consider what the exact objective of the study was, why were the FGs convened, and continue to present those points. As it stands, this manuscript is elevating certain important points, but does not really introduce a framework, nor tools to utilize/apply that framework. I believe this should be considered throughout the revision.

Line 57: It is important to define intersectionality and, of course, important to note its roots in black feminist legal activism

Lines 66-67: The mention of GBV is abrupt and not followed up in the paragraph.

Lines 76-79: Need references to the claims/points made.

Lines 79-82: It would be more helpful if you elaborate on how these power regimes come into play, how intersectionality plays out, how these intertwined layers operate, and explain your points to set the scene for the reader.

Lines 83-84: This sentence is unnecessarily complex. Please rewrite to make your points clear.

Line 85 (and point repeated in Line 132): You introduce SDoH, but you have used political and structural DoH before this point. It is essential to 1. not conflate these concepts with each other, and 2. clearly define whichever one you want to use.

Lines 86-87: While correct, the point on indigenous knowledge seems to be mentioned suddenly and without proper introduction and follow-up.

Lines 88 & 107: Decolonial might be a better word instead of decolonizing/decolonization

Lines 99-100: Use appropriate references to highlight the increasing calls.

Line 126: Needs a reference.

Lines 132-133: This sentence is unnecessarily complex. Please rewrite to make your points clear.

Lines 140-141: Unclear what this sentence adds.

Lines 143-145: “Drawing on constructionist ontology, postmodern epistemology, and interpretivist methodology, we used qualitative methods in this research.”

I recommend starting the methods section with the most important piece of information: this is a qualitative study. Then bring up the abovementioned frameworks to elaborate on how they are applied to qualitative data interpretation.

Line 155: Unclear what you mean by “specific criteria for qualitative research were perceived as appropriate. “Maybe mention the “the following specific criteria.”

Line 153: It is unclear how this section pertains to quality. And what quality? Quality of the overall project? Data quality? Data collection methods quality? Participant’s quality? Their selection criteria quality?

Line 161: Convened instead of conducted.

Line 162: Online meetings platform, Zoom

Line 163: what is the most appropriate composition? You need to elaborate on this appropriateness and its criteria.

Line 167: What do you mean by Majority World, given it has no reference yet is capitalized similarly to a particular name or concept?

Lines 170-172: How was this done and achieved?

Line 182: Elaborate how feminist principles come into play here.

Line 185: A vibrant discussion… is not a scientific term and does not add to this paragraph either.

Data collection section: It is unclear how FG2 and 3 differed. The composition of FGs is not mentioned. It is not clear how many people were in each FG. How the FGs convened in terms of moderation and/or data collection is unclear. What questions, themes, or discussion topics were introduced? How long did each FG meet for? How many times? You need to explicitly mention the gender, level of seniority, region, nationality, educational background, etc., of participants, in aggregate, to paint a clear picture of whose collective contributions led to the following sections while maintaining anonymity.

Figure 1: An arrow moving from deductive/inductive coding to the final code system is missing.

Figure 2: You shouldn’t use abbreviations in figures without introducing them. AAAQ is an abbreviation.

Lines 204-205: Did they? I think this needs to be elaborated.

Lines 212-213: I’m unsure about including this in the positionality and reflexivity statement.

Reflexivity section: It is important to include other co-authors. If they are all FG members, this needs to be explicitly mentioned. If not, their positionality has also influenced the write-up of this research and is worth noting in the section.

Line 230: Figure 3 should come further in the manuscript, perhaps in the discussion alongside recommendations/Table 1, and should be explained and elaborated.

Line 291: Intersectionality

This section can be expanded on much more, given that it is a central theme in the manuscript title as well, but as it stands, it’s not at all informative, nor does it express how intersectionality is relevant to the solutions put forth by the authors.

Line 322: “Feminism itself”. I believe the use of the word “itself” makes it sound conversational. “Feminism” should suffice to convey the message here. If changed, this should be fixed in Figures as well.

Line 332: Objectives and principles section: see overall comments at the beginning.

This section starts by introducing the four principles but moves on not to dive deep into any of the principles; instead, it continues presenting results. It is a very piecemeal section, very long, and the absence of subheadings makes it very difficult to follow and understand.

Line 333: FGD is being used for the first time here.

Line 355: Democracy has not been discussed before here, except for one quote in line 256. If it is one of the four principles, it must be addressed much earlier in the paper.

Lines 367-380: A lot of repetition and not very cohesive. Language should be heavily edited to highlight the most important points and remove repetitions.

Line 387 & 526: UN Sustainable Development Goals 2030 (add reference)

Line 389-390: What do you mean by multiple oppressed individuals? As it stands now, it reads as multiple individuals, i.e., the plurality and number are important. I believe you intend to emphasize individuals experiencing multiple and intersecting layers of oppression.

Line 391: Beginning of what? The policy-making process?

Lines 415-417: Another example of repetitive points that are not really coherent or placed well in the section.

Line 475-476: This paragraph is about academia, yet in the middle of the paragraph you add a very strong point about the health workforce without further elaboration.

Line 482: This claim needs references to some of these feminist economic theories or practices.

Line 548-549: The authors must be very cautious when drawing a conclusion about the robustness and validity of a concept. As this manuscript is submitted for peer review and conducted according to a specific qualitative methodology, validity should not be claimed based on similarities with other published work that have not been conducted following similar or other scientific methodology. The CFFP paper is a desk review type of publication, and the Davies et al. paper is a viewpoint. Both are great additions to the discourse but cannot be used in a point about the validity of a framework introduced in this paper.

Line 566: FGD is used again.

Line 583: Respected or implemented?

Line 600: Terminate? Or transform?

Lines 631-632: The non-response must be elaborated. Does this mean the FGs were convened but participants did not attend? Does the number 11 include all who were invited to FGs, or all who showed up but didn’t contribute to the conversation? Or all who participated and contributed to the conversation? This is a significant point you need to explain.

Lines 657-658: Deeply embedded or informed by?

Line 683: It would be great to elaborate on the recommendations and briefly describe where each recommendation is drawn from - this could happen earlier in the discussion section. For example: A, B, and C, raised by the FG and the literature (add references), led us to make this recommendation: X

Table 1: Recommendations are not actionable, e.g., Adopt more holistic approaches within global health research.

Table 1: Research column, second point: extend, not extent.

6. PLOS authors have the option to publish the peer review history of their article (what does this mean?). If published, this will include your full peer review and any attached files.

**Do you want your identity to be public for this peer review?** For information about this choice, including consent withdrawal, please see our Privacy Policy.

Reviewer #1: No

Reviewer #2: No

---

## [Editor Report · Decision Letter 1]

6 Feb 2024

Towards a Feminist Global Health Policy: power, intersectionality, and transformation

PGPH-D-23-01948R1

Dear Authors,

We are pleased to inform you that your manuscript 'Towards a Feminist Global Health Policy: power, intersectionality, and transformation' has been provisionally accepted for publication in PLOS Global Public Health.

Best regards,

Sreeparna Chattopadhyay, Phd

Academic Editor

Dear Authors,

Thank you for revising the manuscript. I have recommended a provisional yes for publication.

Regards,

Sreeparna